# Cellular Imaging and Time-Domain FLIM Studies of Meso-Tetraphenylporphine Disulfonate as a Photosensitising Agent in 2D and 3D Models

**DOI:** 10.3390/ijms25084222

**Published:** 2024-04-11

**Authors:** Andrea Balukova, Kalliopi Bokea, Paul R. Barber, Simon M. Ameer-Beg, Alexander J. MacRobert, Elnaz Yaghini

**Affiliations:** 1Department of Surgical Biotechnology, Division of Surgery and Interventional Science, University College London, London NW3 2QG, UK; andrea.balukova.16@ucl.ac.uk (A.B.); kalliopi.bokea.19@ucl.ac.uk (K.B.); 2Department of Oncology, UCL Cancer Institute, University College London, London WC1E 6DD, UK; p.barber@ucl.ac.uk; 3Comprehensive Cancer Centre, School of Cancer and Pharmaceutical Sciences, King’s College London, London SE1 9RT, UK; simon.ameer-beg@kcl.ac.uk

**Keywords:** photodynamic therapy, reactive oxygen species, fluorescence lifetime imaging microscopy

## Abstract

Fluorescence lifetime imaging (FLIM) and confocal fluorescence studies of a porphyrin-based photosensitiser (meso-tetraphenylporphine disulfonate: TPPS_2a_) were evaluated in 2D monolayer cultures and 3D compressed collagen constructs of a human ovarian cancer cell line (HEY). TPPS_2a_ is known to be an effective model photosensitiser for both Photodynamic Therapy (PDT) and Photochemical Internalisation (PCI). This microspectrofluorimetric study aimed firstly to investigate the uptake and subcellular localisation of TPPS_2a_, and evaluate the photo-oxidative mechanism using reactive oxygen species (ROS) and lipid peroxidation probes combined with appropriate ROS scavengers. Light-induced intracellular redistribution of TPPS_2a_ was observed, consistent with rupture of endolysosomes where the porphyrin localises. Using the same range of light doses, time-lapse confocal imaging permitted observation of PDT-induced generation of ROS in both 2D and 3D cancer models using fluorescence-based ROS together with specific ROS inhibitors. In addition, the use of red light excitation of the photosensitiser to minimise auto-oxidation of the probes was investigated. In the second part of the study, the photophysical properties of TPPS_2a_ in cells were studied using a time-domain FLIM system with time-correlated single photon counting detection. Owing to the high sensitivity and spatial resolution of this system, we acquired FLIM images that enabled the fluorescence lifetime determination of the porphyrin within the endolysosomal vesicles. Changes in the lifetime dynamics upon prolonged illumination were revealed as the vesicles degraded within the cells.

## 1. Introduction

Photodynamic therapy (PDT) is a minimally invasive procedure which is clinically approved for the treatment of certain types of solid cancers and some non-malignant lesions [1,2,3,4,5,6]. The therapy involves the administration of a photosensitising agent and its photoexcitation by a light source at a specified wavelength corresponding to an absorption band of the photosensitiser (PS), typically at red or near-infrared wavelengths thereby allowing deeper light penetration into tissue. Photoexcitation of the photosensitiser leads to generation of reactive oxygen species (ROS) such as singlet oxygen. These ROS oxidise a range of biomolecules, inducing severe changes in cellular functions ultimately resulting in cell death. Localised and targeted treatment is one of the main advantages of PDT.

A major challenge encountered with many promising nano-sized biotherapeutics and chemotherapeutic agents is that they are subject to cellular uptake via endocytosis and become sequestered within endolysosomes. This can significantly reduce the therapeutic efficacy of such agents since they cannot reach their intended intracellular targets, such as the nucleus, and are also subject to endolysosomal degradation by proteolytic enzymes within lysosomes. Photochemical internalisation (PCI) is derived from PDT and is designed to enhance the intracellular cytosolic delivery and therapeutic efficacy of a range of bioactive agents that are prone to entrapment in endosomes and lysosomes [7]. PCI uses visible light excitation in combination with a co-administered photosensitiser that localises selectively in endolysosomal membranes. The photo-oxidative damage induces the permeabilization of the endolysosomal membranes and the subsequent release of the endolysosomally sequestered agents into the cytosol. PCI has proven to be effective in a wide range of experimental cancer models [8,9,10,11,12], including multidrug-resistant cancer cells, and has shown promising results in a clinical trial of head and neck cancer using the chemotherapeutic agent bleomycin [13], and for cholangiocarcinoma using gemcitabine [14].

A clinically useful common feature of PDT and PCI is that locoregional treatment of disseminated lesions over a wide area is possible using diffuse light irradiation, which is relevant to bladder and ovarian cancer treatment. Ovarian cancer, of which epithelial ovarian cancer (EOC) is the most common type, accounts for 5% of all cancer deaths in women, making it the most lethal gynaecological cancer [15,16]. Unfortunately, many cases of ovarian cancer are diagnosed at a late stage when disseminated disease is already present in the peritoneum. The current standard of care treatment involves surgery to remove the bulk of the tumour, followed by chemotherapy. Most patients initially achieve a complete response after first-line chemotherapy; however, many will relapse requiring further treatment, with an increased risk of chemoresistance. Other treatments have been investigated, including radiotherapy and intraperitoneal chemotherapy instead of standard intravenous chemotherapy following surgery, but with limited success and poor prognosis [17,18,19]. The development of peritoneal carcinomatosis is a major risk factor for these patients which has led to considerable interest in the application of phototherapeutic treatment in the form of either PDT or photoimmunotherapy using targeted PS or PCI, whereby a circumferential light source is placed intra-abdominally so that disseminated malignant nodules in the peritoneum can be treated in parallel [20].

Singlet oxygen, generated via the Type 2 energy transfer mechanism, is highly reactive and exhibits a relatively short lifetime and diffusion distance in the cellular microenvironment, and is one of the key ROS involved in PDT. Cellular damages induced by PDT can also be attributed to the generation of several other ROS, including superoxide that can be produced via an electron-transfer Type 1 mechanism [21,22]. The relative importance of Type 1 and 2 processes is dependent on the cellular oxygenation levels, since the singlet oxygen Type 2 pathway is likely to become less efficient under hypoxic versus normoxic conditions. The localisation and uptake of the photosensitiser inside the cells are also key factors for determining their photodynamic efficacy [23]. Understanding the photochemical, biological, and physical properties of photosensitisers is therefore essential for the optimisation of these treatments.

In this work, we investigated the amphiphilic photosensitiser, meso-tetraphenylporphine disulfonate (TPPS_2a_), in human ovarian cancer HEY cells derived from a peritoneal deposit of a papillary cystadenocarcinoma of the ovary. TPPS_2a_ is amphiphilic as it bears two sulfonate groups each attached to adjacent phenyl rings of the macrocycle. This structure is favourable for PCI, since the sulfonated hydrophilic side of the PS localises to the aqueous-lipid interface of cellular membranes whereas the non-sulfonated hydrophobic macrocycle extends further into the lipid bilayer [24]. TPPS_2a_ was one of the first photosensitisers studied for PCI and its structure is analogous to that of the disulfonated tetraphenyl chlorin (TPCS_2a_, fimaporfin) that is being used in clinical trials of PCI [25]. TPPS_2a_ therefore serves as a suitable surrogate or model photosensitiser for PDT and PCI. We have previously studied the PDT and PCI efficacy of TPPS_2a_ in HEY cells in 2D and 3D spheroid models using saporin as the chemotherapeutic agent [26,27], but did not address the photophysical/chemical properties of this photosensitiser in cells.

Whilst the photophysical/chemical properties of the tetrasulfonated TPPS_4_ derivative have been studied in cells over many years, less is known about the disulfonated TPPS_2a_ derivative, despite its considerable superiority for PDT and PCI [25]. In the first part of the study, the photo-oxidative mechanism by which TPPS_2a_ induces cancer cell death was investigated using ROS detection reagents and free radical scavengers. To complement the ROS studies, fluorescence lifetime imaging microscopy (FLIM) was used to study the photophysical characteristics of the porphyrin in cells [28]. Hitherto, only solution-phase photophysical data have been available for this porphyrin [29,30,31]. Singlet oxygen studies in cells were also carried out using TPPS_2a_. Thus, this study attempts to present a comprehensive microspectrofluorometric approach to investigating the mechanism of PDT-generated ROS and the photophysical properties of this photosensitiser.

The studies were performed in both two-dimensional (2D) monolayer cultures and three-dimensional (3D) in vitro cancer models. 3D cancer models mimic more accurately the in vivo tumour microenvironment by the incorporation of matrix proteins, including collagen and laminin, and the presence of multiple cell types depending on the model’s complexity. Therefore, they recapitulate the missing cell-extracellular matrix (ECM) interactions in 2D models and enable the study of the cell-cell and cell-ECM interactions. Thus, 3D cancer models allow more accurate predictions of treatment responses [32]. The 3D model used in this study employed Type 1 collagen as the matrix component but, unlike standard collagen hydrogels, the matrix density was close to physiological levels through the application of plastic compression [33,34,35]. The resulting dense matrix reduces the rate of oxygen diffusion and replenishment of the oxygen supply to cells. The matrix density is an important factor to consider in our studies, since both PCI and PDT consume molecular oxygen and are aimed at treating solid tumours that are often hypoxic at their core.

## 2. Results

### 2.1. Intracellular Uptake and Localisation of TPPS_2a_

For PDT and PCI, the light-induced cellular damage is directly related to the intracellular location of the photosensitiser. For photosensitisers such as TPPS2a, it is important that they localise preferentially in lysosomes which can sequester macromolecular agents and weak base chemotherapeutics on account of their relative acidity. The incubation times used of up to 24 h correspond to those used for PCI studies of TPPS2a. The cellular uptake and localisation of TPPS2a was assessed by confocal microscopy imaging in ovarian HEY cancer cells in 2D. Cells were incubated with 1 μM TPPS2a for 24 h, and then Lysotracker Green was added for 2 h prior to imaging. Co-incubation of the cells with Lysotracker Green demonstrated the lysosomal uptake of the porphyrin TPPS2a within the cells (Figure 1A–C). The 3D constructs were imaged sequentially to detect fluorescence signals of DAPI and TPPS_2a_. Figure 1D–F clearly demonstrates the fluorescence signal from DAPI (Figure 1D) and TPPS_2a_ (Figure 1E) within the spheroids.

### 2.2. Real-Time Imaging of the Photodynamic Action of TPPS_2a_

We examined the changes in the intracellular TPPS_2a_ fluorescence localisation following its photoactivation in both 2D and 3D in vitro models over time. Since the porphyrin exhibits a punctate intracellular pattern, corresponding principally to endolysosomes, it was anticipated that photo-oxidative damage at these sites should modify the distribution of the porphyrin fluorescence as a result of endolysosomal rupture, as required for PCI-induced cytosolic delivery. In addition, these studies served to demonstrate that the on-stage illumination time used for subsequent ROS generation studies was appropriate. 

For the 2D monolayer model, the cells were treated with 2 μM TPPS_2a_ for 24 h prior to confocal imaging. Cells were illuminated at 405 nm (Figure 2A), and an image was initially captured after 1 min of illumination (2B). A noticeable difference can be observed between image A and B, where fluorescence associated with punctate endolysosomal vesicles is clearly visible in image A (see inset). However, after 1 min of light exposure, the fluorescence became blurred and at longer exposure times (Figure 2C,D) resolution of endolysosomes is no longer apparent, consistent with the degradation of endolysosomal membranes due to photo-oxidative damage. Following 5 min of light exposure, the mean cellular intensity decreased by 24.7% ± 0.2%, indicating photobleaching of the fluorescence. Confocal imaging of the porphyrin fluorescence was attempted in 3D constructs but the resolution was insufficient to detect redistribution effects clearly, although the porphyrin fluorescence was localised to the spheroids within the construct. Photobleaching was also observed as in Figure 2D (see Appendix A). A z-stacked image of a 3D construct stained with DAPI is shown in Appendix A.

### 2.3. Detection of Reactive Oxygen Species Generation in 2D and 3D Cancer Models 

A range of probes was utilised for the detection of reactive oxygen species (ROS) following excitation of TPPS_2a_ in both 2D and 3D cancer models of HEY cells. The production of ROS following the excitation of the TPPS_2a_ was first investigated using the oxidation-sensitive fluorescent probe, 2′,7′-Dichlorofluorescein diacetate (DCFH-DA). Since this fluorescein probe is a general measure of oxidative stress, we also assessed generation of singlet oxygen using the singlet oxygen Sensor Green (SG) assay in combination with the singlet oxygen scavenger, sodium azide (NaN_3_). Superoxide anion production was evaluated using the Dihydroethidium (DHE) probe, while the superoxide dismutase-polyethylene glycol (SOD-PEG) probe was utilised as the superoxide scavenger. Chasing of the cells was employed as for PCI protocols unless otherwise stated.

#### 2.3.1. Dichlorofluorescein Diacetate (DCFH-DA) Assay

Dichlorofluorescein diacetate (DCFH-DA) is a cell-permeable probe that converts to a fluorescent form upon oxidation (4). For imaging, a 405 nm laser was used to excite the TPPS_2a_, while a 488 nm laser was utilised for the DCFH-DA probe (Figure 3). A significant increase in mean cellular fluorescence by 30% ± 3% was observed after 5 min illumination (Figure 3C), as calculated using ROI analysis with ImageJ. Control cells, incubated only with DCFH-DA, showed a significantly lower increase in fluorescence compared to the cells incubated with both DCFH-DA and TPPS_2a_ (Figure 3D–F).

Similar results were obtained with the 3D construct models when the cells were incubated with TPPS_2a_ and DCFH-DA. A mean increase in DCFH-DA fluorescence in the spheroids by 83% ± 21% was observed following light illumination at 405 nm for 5 min (Figure 4A–C). No significant increase in DCFH-DA fluorescence was observed when the cells were incubated only with DCFH-DA (Figure 4D–F).

#### 2.3.2. Detection of Singlet Oxygen Generation

The Sensor Green (SG) probe was investigated for the detection of singlet oxygen (^1^O_2_) since it emits green fluorescence following oxidation by singlet oxygen. This study served to complement the imaging experiments using DCFH-DA which is not a specific singlet oxygen probe. The confocal imaging protocol was the same as for the DCFH-DA experiments. Further studies were conducted with a singlet oxygen scavenger, sodium azide (NaN_3_), co-incubated with SG, to confirm generation of singlet oxygen. For the 2D models, the cells were incubated with 1 μM TPPS_2a_ for 22 h. Afterwards, the cells were washed and 10 μM SG was added for a further 2 h. Similarly, for the 3D models, 2 μM TPPS_2a_ was added for 22 h. Afterwards, the cells were washed and incubated with 20 μM SG for a further 2 h.

For the 2D model, when the cells were illuminated on-stage with the 405 nm laser (5 min), the SG fluorescence in co-treated samples (TPPS_2a_ + SG) increased by 101% ± 15% (Figure 5A–C). The sites exhibiting the highest fluorescence increases are visible as bright green spots (Figure 5C). The punctate pattern mirrors that of the porphyrin fluorescence in Figure 5B. Controls showed that this could not be due to a cross-channel artefact. The SG-only treated samples (Figure 5D–F) were illuminated in the same manner, serving as controls, and a decrease in SG fluorescence was observed. When cells were incubated with the azide (NaN_3_) singlet oxygen quencher (Figure 5G–I), no increase in SG signal was detected. These additional singlet oxygen inhibition experiments confirmed that the increase in SG fluorescence in co-treated samples was indeed due to the generation of singlet oxygen.

In the 3D construct studies, the co-treated samples (TPPS_2a_ + SG) also exhibited an increase in SG fluorescence following 405 nm illumination for 5 min. When the samples were illuminated at 405 nm (Figure 6A,C), the mean SG fluorescence in spheroids increased by 78% ± 15%, indicating generation of singlet oxygen following light illumination. SG-only control samples were illuminated under the same conditions (Figure 6D–F). The SG-only treated samples demonstrated an increase of 6.9% ± 4.4% after 5 min illumination (Figure 6D,F). In contrast to the 2D model, the controls in the 3D model exhibited an increase in SG fluorescence in SG-only treated samples. However, the increase in fluorescence in SG-only treated samples was significantly lower than co-treated 3D models (TTPS_2a_ + SG). With the addition of azide (NaN_3_), the SG fluorescence increased only slightly (by 7.2% ± 0.3%) in the co-treated samples (TTPS_2a_ + SG) (Figure 6G,I). 

#### 2.3.3. Detection of Superoxide

Dihydroethidium (DHE) is widely used for the detection of the superoxide anion. To investigate photo-induced generation of superoxide anions in cells, both 2D and 3D models were co-treated with TPPS_2a_ and DHE. The imaging process was the same as in the SG and DCFH-DA experiments, where an image of DHE was obtained before and after photosensitization with TPPS_2a_, with and without 5 min illumination, and images of TPPS_2a_ were obtained at the beginning and end of the illumination. These experiments were more challenging than the Sensor Green studies, and significant results were only obtained using on-stage illumination with a red laser which is absorbed selectively by the porphyrin and not the DHE probe. 

In the 2D monolayers, no significant changes were observed in DHE fluorescence following 5 min of 405 nm illumination (Appendix A). Likewise for the 3D construct model, 405 nm illumination only showed a small (10%) but not statistically significant increase in DHE fluorescence in spheroids. However, using 638 nm illumination, where the DHE probe has negligible absorption, an increase by 60% ± 18% was observed (Figure 7A–C). In the control sample, the DHE fluorescence (Figure 7D,E) increased only slightly after 5 min illumination of the TPPS_2a_. In order to ascertain whether the increase in DHE fluorescence could be due to PDT-generated superoxide, an inhibition experiment with SOD-PEG was conducted to scavenge the superoxide. In the co-treated (TPPS_2a_ + DHE + SOD-PEG) sample (Figure 7G–I) the DHE fluorescence showed no significant change. The DHE increase observed in cells incubated with TPPS_2a_ + DHE, together with the results of the inhibition experiment (TPPS_2a_ + DHE + SOD-PEG), indicates the presence of that PDT-generated superoxide within the 3D spheroids. 3D spheroids are relatively hypoxic compared to 2D monolayers, thus favouring Type 1 processes. 

### 2.4. FLIM Studies

In the FLIM studies on 2D cell cultures, we investigated the porphyrin lifetime following 24 h incubation using various concentrations (2, 4, and 6 μM) together with control samples without porphyrin. We were able to obtain reproducible FLIM data under all conditions tested, with control samples eliciting much lower intensities. Better resolution can be obtained with 2D samples compared to the 3D construct, where the increased light scattering from the matrix and the thickness of the constructs degrade the resolution. Therefore, we confined the FLIM study to 2D cell cultures. 

Multiexponential fluorescence decays were generally observed in the cells, with the exception of the lowest concentration used of 2 μM, where decays could be fitted using a mean monoexponential lifetime of c. 11 ns. In contrast, at the highest concentration of 6 μM, the decays needed to be fitted using a biexponential function with a shorter minor lifetime component, as determined by the pre-exponential amplitude (A factor).

Figure 8 shows FLIM images of cells (4 μM porphyrin concentration) that demonstrate the high subcellular resolution that we could acquire, showing the punctate endolysosomal fluorescence localization (Figure 8A, top panel). Since these vesicles are the photochemically relevant sites for PCI delivery of endolysosomally-entrapped agents to the cytosol, we studied the fluorescence lifetimes of these sites in further detail and compared lifetimes before and after prolonged on-stage illumination at 520 nm. Lifetime analysis of cells in the initial FLIM image using bi-exponential fitting (Figure 8B,C, top panel) gave a shorter minor component with a lifetime of c. 2 ns. Following on-stage illumination with the FLIM excitation laser, the fluorescence distribution became more diffuse as shown in Figure 8A, bottom panel, and fewer punctate fluorescent vesicles were apparent. The lifetimes were also altered, becoming slightly longer (Figure 8B,C, bottom panel) together with a reduced A factor (A2) compared to that of the main lifetime component (A1). To illustrate this effect, the arrow highlights vesicles which show that the mean lifetime is longer following illumination, as shown by the transition from blue colouration to green/yellow. 

Figure 9 demonstrates a FLIM image and an integrated intensity image (6 μM porphyrin concentration) showing that the shorter lifetime is mainly localised to the punctate vesicles. From the colour-coding (Figure 9B), it is apparent that the fractional lifetime contribution of the short component (as defined in Equation (2)) varies across the vesicles up to about 0.2. 

### 2.5. Singlet Oxygen Detection in 3D Culture

We investigated the detection of singlet oxygen NIR phosphorescence within the 3D spheroids, which were incubated with porphyrin (TPPS_2a_), following excitation at 532 nm. Prior to the experiment, the medium was replaced by D_2_O PBS to improve detection since singlet oxygen undergoes rapid quenching in the presence of H_2_O (the singlet oxygen lifetime in D_2_O is 68 μs compared to 3.3 μs in H_2_O) [36]. A range of incubation conditions were employed, including the effect of adding sodium azide ions, which rapidly quench singlet oxygen, and which served as a positive control. Since sodium azide ions will not penetrate the 3D spheroids immediately, traces were recorded at several time-points after addition.

Figure 10 shows a typical decay trace with a decay lifetime of 50 μs obtained from a 3D sample incubated with the porphyrin (5 μM) for 24 h together with the effect of adding azide (0.5 mM) which significantly quenches the signal (lifetime reduced to 10 μs) and demonstrates that we were able to detect singlet oxygen. Analysis of the residuals demonstrates the goodness of the fitting, with chi-squared values close to unity. Controls without addition of porphyrin showed no observable emission over this timescale. These results were reproducible, but in each case we were only able to observe a relatively long-lived decay. A short-lived initial spike of <5 μs masked any observable rise-time, and its magnitude was smaller in the control samples suggesting it mainly originates from the photosensitiser.

## 3. Discussion

Amphiphilic porphyrins and their chlorin and phthalocyanine analogues have attracted considerable interest for their application in photodynamic therapy and photochemical internalisation. The main aim of this study was to investigate the photoproperties of the amphiphilic porphyrin photosensitiser, meso-tetraphenylporphine disulfonate, TPPS_2a,_ using steady-state microspectrofluorimetric analysis of intracellular porphyrin fluorescence and ROS generation together with time-domain fluorescence lifetime imaging with single photon counting detection. The motivation for this study stems from its application for photochemical internalisation (PCI), which utilises low dose photodynamic therapy for cytosolic delivery of endolysosomally-entrapped agents such as chemotherapeutics or antigens. PCI is currently being developed for both oncological and non-oncological applications, including vaccine delivery [37,38].

The TPPS_2a_ photosensitiser is known to be effective for both PDT and PCI and has an analogous structure to the current clinically used PCI photosensitiser, TPCS_2a_ [25,39,40]. TPPS_2a_ therefore serves as a model photosensitiser for both PDT and PCI. In recent studies, TPPS_2a_ has also been incorporated into nanocarriers for photo-induced enhancement of drug delivery following endocytic uptake [41,42,43].

We have previously studied the PDT and PCI efficacy of this porphyrin in the same ovarian carcinoma HEY cell line in 2D and 3D models using saporin as the chemotherapeutic agent [25,26]. Most in vitro photodynamic therapy (PDT) studies have been carried out in 2D monolayer cell culture models. However, the inherent simplicity of these models cannot accurately simulate the in vivo responses to such therapies. To address this limitation, 3D models have been introduced, which attempt to replicate the 3D arrangement of cancer cells embedded within an extracellular matrix (ECM). Such models can also replicate the hypoxic conditions that are known to prevail in solid tumours. In this study, alongside the 2D monolayer in vitro model, the light-induced generation of reactive oxygen species in 2D and 3D in vitro ovarian cancer models were investigated in 3D compressed collagen spheroids. Time-lapse confocal microscopy combined with on-stage illumination was evaluated to study the fluorescence dynamics of TPPS_2a_ upon light activation, and to understand the mechanism by which this porphyrin induces phototoxicity by ROS generation. 

### 3.1. Intracellular Fluorescence Dynamics of TPPS_2a_ in HEY Cells and ROS Detection

The TPPS_2a_ photosensitiser is an amphiphilic photosensitiser with two negatively charged sulfonate groups located on adjacent phenyl rings of the porphyrin macrocycle. Its structure enables the porphyrin to localise on the aqueous-lipid interface of lipid membranes, whereas the more hydrophobic part of its macrocycle is localised within the lipid bilayer. This is advantageous for PCI, as it enables the photosensitiser to localize in endolysosomal membranes wherein drugs are entrapped [24]. The intracellular localization of TPPS_2a_ was observed in the HEY cells following incubation for 24 h (Figure 1). Co-incubation of cells with LysoTracker Green demonstrated the localisation of TPPS_2a_ within lysosomes.

Porphyrin photosensitisers are generally considered to exert their phototoxic activity via Type 1 and 2 mechanisms. An example of the Type 1 mechanism is where an electron is transferred between the excited photosensitiser and a biological substrate, forming radicals and radical ions. These radicals interact with the available oxygen molecules and produce oxygenated products, such as the superoxide ion (O_2_^−^). Direct electron transfer to oxygen to form superoxide is not thought to be significant. Alternately, in the Type 2 mechanism, resonant energy-transfer occurs from the excited photosensitiser triplet state to convert ground state oxygen (^3^O_2_) to the highly reactive short-lived singlet oxygen species (^1^O_2_). Singlet oxygen is considered to be a principal mediator of the phototoxic effect of PDT; however, products of the type 1 reaction, such as superoxide anions, may also be involved, particularly under hypoxic conditions [21]. Since these ROS will target different parts within the cells, there may be an advantage in terms of synergistic activity and oxygen dependence. As the photosensitiser is mainly localized in endolysosomal vesicles, and specifically in the vesicle membranes owing to its amphiphilic properties, generation of reactive oxygen species will lead to membrane oxidative damage, permeabilization, and ultimately rupture. This effect is illustrated in Figure 2 where light-induced fluorescence redistribution is evident from punctate vesicles to a diffuse distribution.

We can detect the generation of ROS using chemical probes that change their fluorescence properties after oxidation. The DCFH-DA probe is commonly used to detect oxidative stress but is not specific to singlet oxygen. To detect singlet oxygen specifically, we investigated the use of the Sensor Green singlet oxygen probe with and without the addition of sodium azide, which is an efficient physical quencher of singlet oxygen. For superoxide detection, we used the DHE probe. However, since the DHE probe is not completely specific to superoxide, we employed a cell-permeable superoxide inhibitor, superoxide dismutase polyethylene-glycol (SOD-PEG), using a concentration previously employed [44].

The intracellular generation of ROS due to the photoactivation of TPPS_2a_ was confirmed by confocal microscopy using these probes and on-stage illumination, mainly at a wavelength of 405 nm. The DCFH-DA probe is non-fluorescent until it undergoes oxidation. In both the 2D monolayer model (Figure 3) and the 3D spheroid construct model (Figure 4), the DCFH-DA fluorescence increased significantly following 5 min illumination vs. controls, which is consistent with light-induced intracellular ROS generation. 

To specifically detect singlet oxygen, the singlet oxygen Sensor Green (SG) probe was used in both 2D and 3D models, with and without a singlet oxygen quencher (NaN_3_). The probe was incubated with cells without serum protein present during the short ‘chasing’ period prior to illumination, according to the standard PCI protocol. This ensures better cellular uptake of the probe, uptake of which would otherwise be limited by binding to the serum proteins in the extracellular medium. The ROS detection studies in the 2D model showed an increase in fluorescence for both the 405 nm and 552 nm excitation. The inhibition experiment using sodium azide ions as the singlet oxygen quencher showed a significantly lower increase in fluorescence, which is consistent with the fluorescence increase being attributable to singlet oxygen (Figure 5). Interestingly, we noted that the brightest sites of fluorescence increase corresponded to small spots, which may be related to the initial endolysosomal distribution of the photosensitiser. In the 3D model, the SG fluorescence increased significantly as well, and azide inhibition confirmed the involvement of singlet oxygen (Figure 6). Overall, the singlet oxygen detection and inhibition experiments in both 2D and 3D models confirmed that one of the phototoxic mechanisms of light-activated TPPS_2a_ is through the generation of singlet oxygen. 

We also attempted to detect superoxide generation using dihydroethidium bromide (DHE) together with the superoxide inhibitor, SOD-PEG. We reasoned that since the 3D spheroid model is known to be hypoxic, this should promote the Type 1 mechanism in comparison to normoxic conditions where the Type 2 mechanism would be dominant (Figure 7). Compared to the SG fluorescence increase in cells, the DHE fluorescence increases were marginal, and were significant only in 3D model experiments. Although the DHE probe employed is not completely specific to superoxide, we did employ a cell-permeable form of superoxide dismutase as the inhibitor which should have high specificity for superoxide. Importantly, co-incubation with the SOD-PEG scavenger abrogated the fluorescence increase. However, we had to use red illumination at 638 nm instead of 405 nm, which did not produce significant increases in fluorescence. Since DHE also absorbs at 405 nm in contrast to 638 nm, it is possible that auto-oxidation of the dye leading to artefactual fluorescence masked any fluorescence increase. In addition, ROS generation by endogenous chromophores is less likely using red light than 405 nm. We therefore suggest that the use of red illumination in ROS probe imaging using photosensitisers should be more widely adopted, where possible. Since the 3D models are relatively hypoxic compared to monolayers, this observation would bear out the interchange between Type 1 and 2 mechanisms and may warrant further investigation. The indication that superoxide is generated was unexpected, however, and it is possible that photocatalytic Type 1 interaction of photoexcited porphyrin with NADH can mediate enhanced levels of superoxide generation. Dimerisation of the porphyrin, as discussed below in relation to the FLIM studies, may also promote a Type 1 mechanism. In our study, we also assumed that superoxide is the active species, rather than peroxyl radicals, for example. Further PDT and PCI studies under hypoxic conditions, including inhibition of cellular toxicity using SOD-PEG or another suitable probe/inhibitor combination, may provide useful information and confirmation in this regard. Finally, we note that the mechanism by which light-activated photosensitisers induce membrane permeabilization may not simply rely on the generation of lipid hydroperoxides via a type 2 singlet oxygen mechanism. Bacellar and Baptista have recently proposed a more detailed mechanism that relies on direct interaction, or a ‘contact mechanism’ between the photosensitiser triplet state and oxidized lipids [45], and it is therefore possible that this mechanism may play a role in PCI [45].

### 3.2. FLIM and Singlet Oxygen Detection

To gain a fuller understanding of the photoproperties of TPPS_2a_ in cells, we investigated the photophysical properties of TPPS_2a_ in cells using time-domain fluorescence lifetime imaging. The photophysical properties of TPPS_2a_ are relatively well studied in model systems, including the effect of varying the pH in aqueous solution, solvent polarity, and uptake into micelles, all relevant factors to consider when extrapolating to cellular systems. This dataset, therefore, proved valuable for the interpretation of fluorescence lifetime decays in cells. In addition, TPPS_2a_ is structurally very similar to the disulfonated tetraphenyl chlorin (TPCS_2a_) that is used clinically for photochemical internalisation, and thus serves as a protype PCI photosensitiser. We previously studied the phototoxicity of TPPS_2a_ and its efficacy for PCI in the same cell line in 2D and 3D models [26], and others have demonstrated that the mechanism of phototoxicity is based on lysosomal damage leading to autophagy-associated cell death [24,41], which is likely to stem from the endolysosomal membrane localisation of TPPS_2a_ [23,46]. Excitation of the porphyrin results in generation of the long-lived excited triplet state and reactive oxygen species, principally singlet oxygen, with a high quantum yield of 0.7 in methanol [41].

Although cellular uptake and localisation of this porphyrin have been previously studied, relatively little is known about its photophysical properties in cells. FLIM lifetime analysis showed that the porphyrin fluorescence in cells generally exhibited multiexponential decays, and that good fits could be obtained using biexponential fitting (Figure 8). The major component of c. 12 ns is close to published model system studies in simple solvents: e.g., 11.1 for the aprotic solvents ethylene glycol and propan-2-ol [30], which are appropriate comparisons since the porphyrin is predominantly localised within lipid membranes. A minor shorter lifetime component of c. 2 ns was observed as well, which was mainly confined to the bright, punctate vesicles corresponding to the endolysosomes where this porphyrin localises. We did observe some limited interdependence of the minor and major lifetime components when different concentrations were employed. For the lowest concentration of 2 μM, lifetimes could be adequately fitted using monoexponential decays. We also examined the effect of prolonged on-stage illumination by the FLIM excitation laser at 520 nm. As shown in Figure 8, the punctate fluorescence pattern became indistinct, which is attributed to light-induced photo-oxidative rupture of the endolysosomal lipid membranes, thereby releasing the porphyrin into the cytosol and enabling it to bind with other components. Accordingly, the lifetimes observed changed as the porphyrin microenvironment was altered, and the contribution of the shorter lifetime component was markedly reduced based on relative A factors (normalised values). In Figure 9, further analysis revealed the spatial distribution of the fractional contribution of the shorter lifetime component, and the highest values for the fractional contribution were observed in the bright vesicles.

We attribute the shorter lifetime component to aggregation of the porphyrin (e.g., dimers and higher order aggregates) which is known to increase the non-radiative decay rate of photoexcited porphyrins, thereby shortening the fluorescence lifetime. Owing to the presence of the two adjacent sulfonate groups, aggregation of TPPS_2a_ in its ionised deprotonated form leads to a face-to-face alignment of the macrocycles to form H-aggregates [47]. Aggregation of the clinically used disulfonated chlorin analogue (TPCS_2a_) is likely to form the same type of aggregates [40]. 

Lilletvedt et al. attributed shorter lifetimes of c. 2ns and 0.5 ns, observed in solution phase studies of TPPS_2a_ using non-polar solvents and methanol, to aggregation of TPPS_2a_ [30]. Since aggregation is concentration-dependent, the presence of the shorter lifetime in the endolysosomes, where the porphyrin localises, would be consistent with this argument since a larger degree of aggregation-induced quenching and lifetime reduction should therefore occur due to the higher porphyrin concentrations present. Release of the porphyrin will result in dilution, which should abrogate aggregation, thereby leading to a longer lifetime, as shown in Figure 8. This conclusion is also consistent with the lower contribution of the shorter lifetime component using the lowest porphyrin incubation concentration at 2 μM. Although we assign the short lifetime component to the aggregated sensitiser, albeit the minor component, we note that aggregation is nevertheless likely to impair photodynamic efficacy [48] since the self-quenching of the photoexcited porphyrin will reduce the probability of triplet state mediated Type 1 and 2 processes. 

There is one previous study of FLIM of TPPS_2a_ that we are aware of [41] where TPPS_2a_, encapsulated in polymer nanoparticles, was incubated with HeLa cells. A different raster scanning FLIM system (Picoquant MicroTime 200) was employed in that study. They observed a mean lifetime of ~10 ns for the encapsulated TPPS_2a_ in cells and noted that this value was about 2 ns shorter than that found for aqueous suspension. It was suggested that this shorter lifetime might indicate partial intracellular release of the porphyrin from the nanoparticles, but no further lifetime analysis was presented. We have previously used FLIM to study the TPCS_2a_ chlorin analogue in cells using two-photon excitation at c. 800 nm, and obtained a mean lifetime of 6 ns, which was shorter than that observed in methanol at 8.5 ns [49]. Multiphoton excitation has also been used for PCI delivery, albeit with a different porphyrin sensitizer [50]. However, the spatial resolution obtained in our previous FLIM study on TPCS_2a_ was far inferior to that obtained herein, which demonstrates the recent advances that have been made in the FLIM technique.

Our data can be compared with previous fluorescence lifetime studies of the tetrasulfonated TPPS_4_ derivative in cell suspension and model solutions [51,52]. This derivative has been more widely investigated for PDT than the disulfonated derivative, mainly due to the ease of synthesis of tetra-substituted porphyrins. In studies of the dependence of lifetime in aqueous solution as a function of the pH, Schneckenburger et al. observed multiexponential decays of TPPS_4_ with a short lifetime of 1.3 ns which they suggested might originate from aggregates, together with a lifetime of 10.6 ns and an intermediate lifetime of 4.8 ns. They attributed the intermediate lifetime to the dicationic moiety where the central amino nitrogens are protonated at acidic pH [51]. This conclusion is in agreement with results from an earlier study by Harriman and Richoux [53]. Using time-gated imaging with an intensifier, they observed multiexponential decays in RR1022 sarcoma cells, although this technique only provided an approximate estimate of the two lifetimes of c. 11 ns and a minor one of < 3ns. In a more recent TCSPC study on suspensions of human fibroblasts, Jiménez-Banzo et al. observed biexponential decays with a major component of c. 14 ns and a minor component of 3.7 ns [53]. As with the earlier study by Schneckenburger et al. [51], they attributed the shorter lifetime to the protonated dicationic moiety. Protonation is likely to occur for the TPPS_4_ derivative in cells since it resides within the central aqueous compartment of endolysosomes following adsorptive endocytosis. Since the pH of lysosomes is acidic at c. pH 5, a fraction of the tetrasulfonated porphyrin becomes protonated within cells; therefore, one would expect to observe a short lifetime component of a few ns corresponding to the dicationic porphyrin moiety.

The fluorescence decay and emission spectra of the TPPS_2a_ derivative are also known to be pH dependent, and in aqueous solution the protonated dicationic moiety exhibits a pKA of ~4 [54,55]. However, in cells, the same scenario will not necessarily apply to the disulfonated derivative following endocytosis because, in contrast to the tetrasulfonated derivative, the disulfonated porphyrin is amphiphilic and resides predominantly in the endolysosomal membrane instead of the central aqueous compartment. Owing to its membrane localisation, the central imino nitrogens of the TPPS_2_a porphyrin are less likely to become protonated. In support of this argument, Nardo et al. observed a TPPS_2a_ fluorescence lifetime in aqueous Pluronic 127 micellar solution of 11.7 ns with no apparent pH dependence even as low as pH 2.9 [31]. 

We attempted to detect singlet oxygen phosphorescence from 3D spheroids sensitised with TPPS_2a_, but we were only able to detect singlet oxygen following H_2_O/D_2_O exchange (Figure 10). We observed a relatively long-lived lifetime characteristic of singlet oxygen in D_2_O, albeit slightly shorter at 50 μs. We could not determine any risetime due to an intense short-lived initial spike which we ascribe mainly to the residual long wavelength tail of porphyrin fluorescence. The relatively long lifetime suggests that we were mainly detecting singlet oxygen outside the cells, since singlet oxygen can be scavenged rapidly within cells reducing its lifetime to c. 1 μs [56]. Our data are in accordance with a previous study by Alemany-Ribes et al. who performed similar experiments on 3D spheroid models of human fibroblasts incubated with the cationic TMPyP4 porphyrin (5,10,15,20-tetrakis(N-methyl-4-pyridyl)-21H,23H-porphine) in a peptide-based extracellular matrix [57]. They observed a long singlet oxygen lifetime of 54 μs comparable to that observed here, and suggested that the reduced value compared to D_2_O was due to quenching by the extracellular matrix. Sulfonated tetraphenyl porphyrins also emit weak phosphorescence at c. 900 nm [58,59], therefore, future studies of the sensitiser phosphorescence in cells could provide further useful information, including the rate of triplet state decay which will be oxygen dependent. These porphyrins can also exhibit delayed fluorescence [60], although the intensity is very low compared to prompt fluorescence.

In summary, we demonstrated that detailed intracellular fluorescence lifetime data can be acquired for this porphyrin photosensitiser using the FLIM technique at high resolution so that we were able to resolve subcellular organelles, which represents an advance on previous studies of photosensitisers. The results of steady-state imaging on the intracellular porphyrin localisation and light-induced relocalisation are also consistent with the data provided by fluorescence lifetime imaging and demonstrate the effect of illumination on the endolysosomal vesicles and aggregation state of the porphyrin. In the FLIM study, the porphyrin was excited at 520 nm, but the use of a blue laser near 400 nm, corresponding to the intense Soret absorption band, would enable a ten-fold increase in excitation efficiency and shorter acquisition times and/or the use of lower concentrations. We also note that porphyrins have relatively low fluorescence quantum yields of c. 0.1 compared to many other photosensitisers, therefore the technique has considerable potential for future photophysical studies using FLIM. 

## 4. Materials and Methods

### 4.1. Cell Culture

The human ovarian carcinoma cell line (HEY) was obtained from the American Type Culture Collection (ATCC) and was cultured in DMEM/F-12 medium (Sigma-Aldrich-Aldrich, Dorset, UK). The cell culture medium was supplemented with 10% FBS (Sigma-Aldrich-Aldrich) and 1% penicillin/streptomycin (Sigma-Aldrich-Aldrich).

### 4.2. Manufacture of 3D Cancer Constructs

The RAFT 3D culture systems (Lonza, Slough, UK) protocol was followed to manufacture the 3D in vitro compressed cancer constructs in 96-well plates. To create a compressed 3D construct, a hydrogel containing the cells was prepared and subjected to compression by fluid extraction absorbers, producing 3D constructs of around 200 μm thickness. To prepare the hydrogel, 10X Medium Essential Medium (Gibco, Scotland, UK) and Rat Tail Collagen Type 1 (First Link UK Ltd., Custom Bio-Reagents, Birmingham, UK) were initially mixed. A neutralizing solution of NaOH and HEPES buffer (Gibco) was then added to the mixture. Cells were added to the collagen mixture at a concentration of 50,000 cells per well. The cell mixture was further seeded into a 96-well plate at a volume of 240 μL per well, followed by incubation for 15 min (at 37 °C, 5% CO_2_) to set the hydrogel. Afterwards, the constructs were subjected to plastic compression using RAFT absorbers for 15 min (at room temperature). Upon removal of the absorbers, culture medium was added, and the 3D cancer constructs were incubated at 37 °C, 5% CO_2_. 

### 4.3. Confocal Microscopy and Cell Preparation

Confocal microscopy was performed on 2D cancer monolayers and 3D cancer constructs using a Leica SP8 upright confocal microscope (Leica Biosystems, Newcastle, UK) to investigate the intracellular uptake and localisation of the porphyrin, and on-stage illumination for the investigation of the photo-oxidative mechanism using ROS detection reagents and free radical scavengers. The system is equipped with lasers operating at 405, 488, 552, and 636 nm, which were used for fluorescence excitation and, with longer exposure times, for on-stage illumination to induce intracellular ROS generation. 

For 2D monolayer studies, cells were seeded at 3000 to 5000 cells/well in 35 mm FluoroDish™ sterile culture dishes with a central coverslip base (World Precision Instruments, Sarasota, FL, USA) in 200 μL supplemented medium. The porphyrin TPPS_2a_ (Frontier Scientific Inc., Carnforth, UK) was added to the cells at 24 h post-seeding and then incubated from 4 to 24 h. To minimise the presence of the porphyrin in the extracellular membrane, as required for PCI, chasing was also employed, whereby after incubation for 20 h with the porphyrin, cells were thoroughly washed with PBS and then fresh medium without the porphyrin was added for a further 4 h. For co-treated samples and controls, ROS probe reagents were added after the washing step for up to 2 h incubation prior to imaging. Prior to imaging, the cell medium was removed, cells were washed, and fresh phenol red-free medium was added. 

To image the 3D constructs, the constructs were treated with TPPS_2a_ on day 7 post-seeding by which time spheroids were typically 50–100 μm in diameter. The constructs were then incubated for 22 h with the porphyrin, after which they were washed with fresh medium. For co-treated samples, ROS probe reagents (Section 2.4) were added at the washing step for up to 2 h incubation. Prior to imaging, the cell medium was replaced with fresh medium without the probe reagents. The constructs were removed from the 96-well plate and mounted on glass slides with a cover slip using fresh phenol red-free medium. Formalin-fixed samples were mounted for imaging in the same manner. 

#### Intracellular Uptake and Localisation

Confocal laser fluorescence microscopy was used to assess the uptake and localisation of the porphyrin. To visualize lysosomes, cells were treated with 100 nM Lysotracker Green DND-26 (Thermo Fisher Scientific, Hemel Hempstead, UK). The porphyrin was imaged using 405 nm excitation and over the 630–670 nm emission range for fluorescence detection. Lysotracker Green was imaged with 488 nm excitation and detection at 510–535 nm. In selected studies to highlight the nucleus, the cells were also stained with 10 μM DAPI (Thermo Fisher Scientific, Hemel Hempstead, UK) for 30 min and washed with PBS. DAPI fluorescence was excited at 405 nm and detected in the 414–542 nm emission range.

### 4.4. Detection and Inhibition of Reactive Oxygen Species (ROS) Generation

For the ROS imaging studies, the following fluorescence-based probes were employed: 2′,7′-Dichlorodihydrofluorescin diacetate (DCFH-DA, Sigma-Aldrich-Aldrich), Sensor Green (SG, Thermo Fisher Scientific, Dorset, UK), and Dihydroethidium bromide (DHE, Thermo Fisher). For ROS inhibition, 25 mM and 50 mM sodium azide (Sigma-Aldrich), and Superoxide dismutase-polyethylene glycol at 200 units/mg protein (Sigma-Aldrich), were investigated as scavengers of singlet oxygen and superoxide, respectively. For ROS imaging using DCFH-DA, excitation was carried out at 488 nm with detection at 510–530 nm. For Sensor Green, fluorescence excitation was carried out at 488 nm and detection at 510–550 nm. For DHE, excitation was at 488 nm with detection at 580–610 nm, which did not overlap with the porphyrin detection channel. For the experiments where two separate fluorescent agents were employed, care was taken to minimise any cross-channel interference between excitation/detection channels as confirmed using appropriate controls.

The production of ROS following light illumination of the porphyrins was investigated in 2D and 3D constructs using 5 μM DCFH-DA. The generation of singlet oxygen in cells was assessed using 20 μM of the singlet oxygen probe, Sensor Green, and production of superoxide was evaluated with 1 μM DHE. Cells were incubated with the relevant ROS inhibitor for up to 2 h and washed prior to imaging. All ROS inhibitors were prepared and added to cells with serum-free medium, and all co-treated samples were washed with serum-free medium. This is particularly important for Sensor Green, since binding to serum proteins can inhibit cell uptake. The residual presence of esterases in serum can also confound the use of the cell-permeable esterified DCFH-DA probe.

For the cells that were co-treated with the porphyrin and DCFH-DA, 488 nm excitation was used to detect the oxidised green fluorescent photoproduct (dichlorofluorescein), and 405 nm excitation was used to detect the porphyrin fluorescence. The samples were scanned sequentially with an image initially taken with the 488 nm laser, where the porphyrin only weakly absorbs, to show the background fluorescence. Then, the cells were illuminated with a 405 nm laser with an incident power of <1 mW (where the fluorescein derivative only absorbs very weakly) for up to 300 s, at the end of which a fluorescence image was recorded of the porphyrin. Finally, an image using 488 nm excitation was taken to detect the change in the fluorescein fluorescence following illumination of the porphyrin. The same protocol was employed for the Sensor Green and DHE probes using the aforementioned detection wavelengths for these probes. Illumination of the cells was also attempted using the 552 nm or 638 nm lasers instead of the 405 nm laser to excite the porphyrin. 

### 4.5. Fluorescence Lifetime Imaging Microscopy (FLIM) and Cell Preparation

Laser scanning fluorescence lifetime imaging of the porphyrin in cells was carried out using a Leica Stellaris Falcon (Leica Microsystems (UK) Ltd., Milton Keynes, UK). Images at 512 × 512 pixel resolution (185 × 185 μm) were then generated and processed to generate the time-resolved FLIM data. Laser excitation was carried out at 520 nm supplied by a super-continuum laser at an 80 MHz repetition rate and c. 1 ps pulse duration. The fluorescence was collected within the 640–680 nm wavelength range. Steady-state fluorescence images were also recorded using the total photon counts. A 63X oil (1.4 NA) objective was used for all imaging, giving a 0.9 μm optical section thickness.

Fluorescence TCSPC data at each pixel were integrated for 30 integrations, and 2% excitation power (7 μW into the objective), to build up sufficient photon counts for statistical analysis. Photon arrival times were integrated into 128 time-bins spanning 12.4 ns. All fitting of the FLIM data was performed in the TRI2 analysis program, which has been previously documented [61p]. The FLIM images/data are not reconvolved since the instrument response function is very short (<<1 ns). The FLIM images/data are not reconvolved since the instrument response function is very short (<<1 ns) which therefore eliminated any fitting artefact. 

Bi-exponential fits to the time-resolved data were obtained using the model:I = Z + A1 exp(−t/tau1) + A2 exp(−t/tau2)(1)

In Equation (1), I represents intensity, A1 and A2 are amplitudes in units of photon counts, t represents time in ns, and tau1 and tau2 are the fitted lifetimes (tau1 is always assigned the longer lifetime). The background Z was fixed to zero in all fitting, because the measured lifetimes were expected to be near the 12.4 ns total measured period. Average lifetimes were determined with a mono-exponential model by setting A2 = 0. Global bi-exponential fitting was performed as previously described [61], which fixed tau1 and tau2 across the image while allowing A1 and A2 to vary. Fractional lifetime contribution of the short component was calculated using the equation:f2 = A2.tau2/(A1.tau1 + A2.tau2)(2)

Cells were plated into chambered 4-well plates with thin glass bases (Ibidi GmbH, Gräfelfing, Germany) and mounted on the inverted microscope stage. For the 2D monolayer imaging, 8000 cells per well were seeded in these plates in cell medium prepared as described above. The cells were then treated with the porphyrin at 24 h post-seeding for 24 h incubation. At least 3 wells were prepared for each concentration or incubation time. All samples were washed in porphyrin-free medium prior to imaging.

### 4.6. Singlet Oxygen Measurement

Singlet oxygen phosphorescence at 1270 nm was detected from photosensitised 3D constructs mounted in a 1 cm quartz cuvette containing a solution of 0.01M PBS in deuterated water. The 3D constructs were prepared as mentioned above and grown for 7 days prior to the experiment. The planar 3D constructs (approx. 4 mm diameter and 0.2 mm thickness) were mounted on a circular wire support and then positioned within the cuvette. The plane of the constructs was aligned vertically at approximately 45 degrees to the incident horizonal laser beam, and the height of the wire support within the cuvette was adjusted so that the laser beam was incident on the central part of the constructs. The central site of the construct was also positioned to ensure optimal luminescence detection by the collection lens of the detection system described below. Concentrated aqueous solutions of sodium azide were added to the cuvette at a final concentration of 0.5 mM to demonstrate suppression of the singlet oxygen signal after periods of up to 30 min, thereby ensuring sufficient azide diffusion.

For time-resolved photon counting detection in the near-IR, a thermoelectrically cooled photomultiplier (model H10330-45, Hamamatsu Photonics Ltd., Hertfordshire, UK) was used, and the emission was collected via a lens mounted orthogonally to the laser beam in combination with a long-pass and a band-pass filter centred at 1270 nm (Interferenzoptik Electronik GmbH, Nabburg, Germany). Excitation was carried out using a pulsed 532 nm Nd:YAG laser (3 kHz repetition rate and 3 ns pulse length, Lumanova GmbH, Hannover, Germany) with the beam axis aligned orthogonally to the collection optics, and a fast photodiode (Becker-Hickl, Berlin, Germany) was used to synchronize the laser pulse with the photon counting system. Calibrated neutral density filters were used to attenuate the laser power. The photon-counting detection equipment consisted of a multiscaler board (MSA-300, Becker-Hickl, Berlin, Germany) and a pre-amplifier (Becker-Hickl, Berlin, Germany). Integrated time-resolved phosphorescence traces were analysed using FluoFit software (PicoQuant GmbH, Berlin, Germany) to extract the singlet oxygen decay lifetimes and amplitudes using tail fitting analysis at c. 5 μs after the initial intense spike that was observed, which was attributed to residual fluorescence and scattering. The residuals for the fitted lifetimes were also plotted to determine the quality of the fit.

### 4.7. Image Analysis

All images of 2D cancer and 3D cancer models were acquired by confocal microscopy and analysed with Image J and Fiji software (NIH, USA, National Institutes of Health (NIH), Bethesda, MD, USA). Regions of interest (ROIs) were selected, and values for mean grey value (intensity) and integrated density (total amount of fluorescence) were obtained and analysed. In uptake studies for TPPS_2a_ in 2D monolayer cultures, changes in fluorescence intensity were analysed for different time points after laser illumination with confocal microscopy. For 3D cancer constructs, Z-stack imaging in real time was performed to obtain a 3D image of the constructs (see Appendix A). Time-lapse images of 2D and 3D constructs were analysed with MATLAB for creating false-colour intensity maps and ratio images. ImageJ software (NIH open source) was used to analyse images and obtain mean intensities for delineated regions of interest (ROI).

### 4.8. Statistical Analysis

Unless otherwise indicated, all experiments were repeated in triplicate. Results are presented as mean values +/− standard deviation (SD), and data were analysed using one-way ANOVA with appropriate post-hoc analysis. Significance was set at *p* < 0.05.

## 5. Conclusions

Overall, we can conclude that the photophysical properties of TPPS_2a_ in cells are not markedly perturbed and that the fluorescence lifetime behaviour is consistent with that observed in model systems where both porphyrin monomers and aggregates are present. This conclusion is supported by the high resolution achieved in confocal FLIM studies that enabled acquisition of lifetime dynamics in intracellular vesicles in which the porphyrin was predominantly localised. The time-lapse confocal imaging methodology in both steady-state and FLIM studies also provides real-time data on photodynamic action of the photosensitiser, and combining this imaging approach with a 3D cancer model will be a valuable tool for gaining insight into the PDT and PCI mechanisms in vivo. The microspectrofluorimetric strategies may also be useful for mechanistic studies of more recently developed approaches that harness the PCI concept, such as nanoparticles which combine both the photosensitiser and the biological agent.

## Figures and Tables

**Figure 1 ijms-25-04222-f001:**
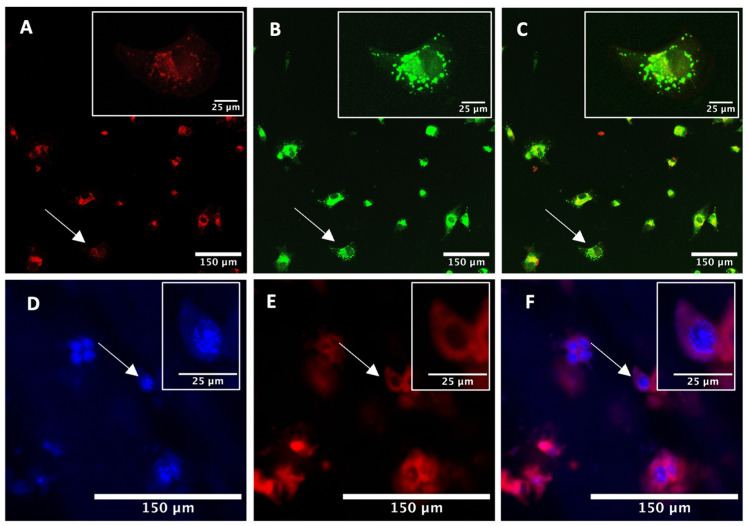
Confocal images of HEY cells in 2D monolayer models (**A**–**C**) and 3D construct models (**D**–**F**). (**A**–**C**): Cells were co-treated with 1 μM porphyrin TPPS_2a_ for 24 h and 100 nM Lysotracker Green. (**A**) TPPS_2a_ fluorescence using 405 nm excitation; (**B**) Lysotracker Green fluorescence using 488 nm excitation; (**C**) represents an overlay image of (**A**,**B**). The scale bar presented in each image is 50 μm. (**D**–**F**): 3D constructs were treated with 1 μM porphyrin TPPS_2a_ for 24 h. Prior to imaging, the constructs were stained with 10 μM DAPI (blue) and incubated for 30 min. The porphyrin TPPS_2a_ (**E**) was imaged in the same manner as in the 2D model, and (**F**) is an overlay of (**D**,**E**). The scale bar presented in each image is 150 μm. Scale bar for the magnified images in insets highlighted by arrows is 25 μm.

**Figure 2 ijms-25-04222-f002:**
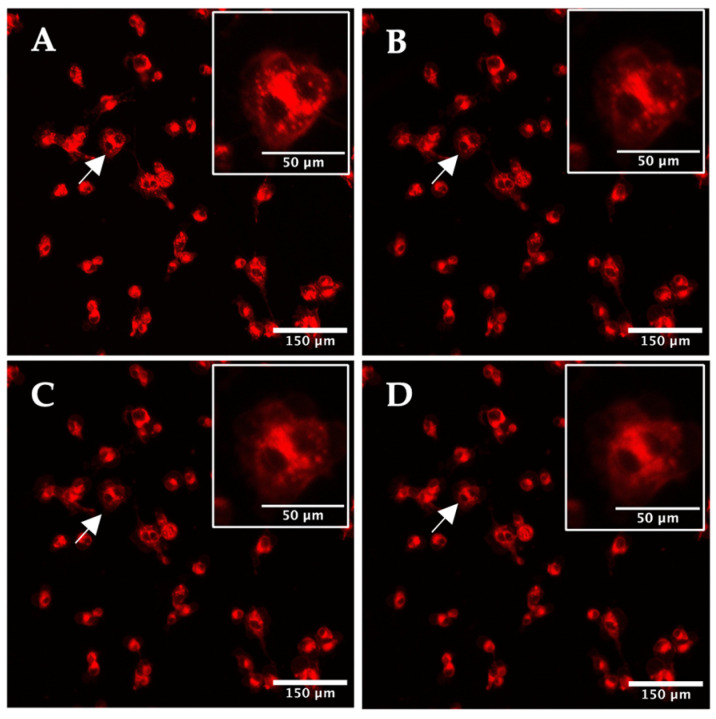
Time-lapse confocal imaging of HEY 2D monolayer culture. Cells were treated with 2 μM TPPS_2a_ for 24 h prior to imaging. Cells were initially illuminated on-stage using the 405 nm laser at a low power setting (10%) and an image recorded (**A**), then the laser power was increased to 80%, and images were taken at 1 min (**B**), 3 min (**C**), and 5 min illumination (**D**). The inset in the top right corner shows an expanded view of the cell highlighted by the arrow to demonstrate intracellular fluorescence redistribution. The scale bar presented in each image is 150 μm. Scale bar for the magnified images in insets is 50 μm.

**Figure 3 ijms-25-04222-f003:**
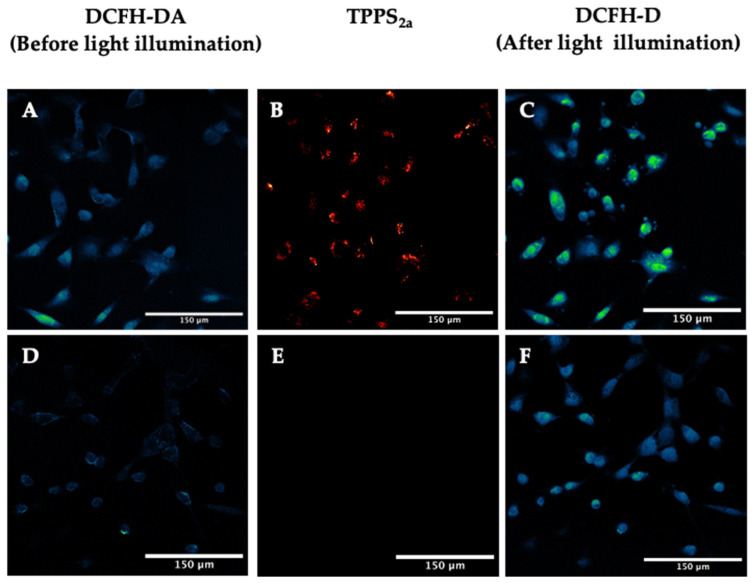
Detection of reactive oxygen species using DCFH-DA assay in 2D monolayers. (**A**–**C**): HEY cells were incubated with 1 μM TPPS_2a_ for 22 h. Prior to imaging, 5 μM DCFH-DA was added for 2 h to the cells. (**A**) DCFH-DA fluorescence before illumination; (**B**) TPPS_2a_ fluorescence; (**C**) DCFH-DA fluorescence after 5 min light exposure. TPPS2a images are shown in red, fluorescein in false colour (blue, low; green, high). (**D**–**F**): HEY cells incubated with 5 μM DCFH-DA for 2 h (without TPPS_2a_). (**D**) DCFH-DA fluorescence before illumination; (**E**) control image showing no TPPS_2a_ fluorescence; (**F**) DCFH-DA fluorescence after 5 min illumination. The scale bar presented in each image is 150 μm.

**Figure 4 ijms-25-04222-f004:**
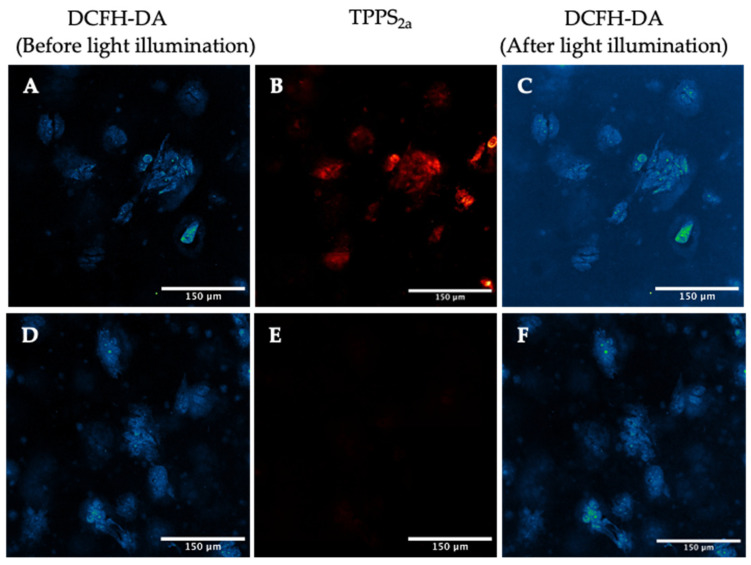
Detection of reactive oxygen species using DCFH-DA assay in 3D spheroid constructs. (**A**–**C**). TPPS2a images are shown in red, fluorescein in false colour. HEY cells were incubated with 1 μM TPPS_2a_ for 22 h. Prior to imaging, 5 μM DCFH-DA was added into the cells. (**A**) DCFH-DA fluorescence before 405 nm illumination; (**B**) TPPS_2a_ fluorescence; (**C**) DCFH-DA fluorescence after illumination. (**D**–**F**): HEY cells incubated with 5 μM DCFH-DA (without TPPS2a). (**D**) DCFH-DA fluorescence before illumination; (**E**) control image showing no TPPS_2a_ fluorescence; (**F**) DCFH-DA fluorescence after illumination. The scale bar presented in each image is 150 μm.

**Figure 5 ijms-25-04222-f005:**
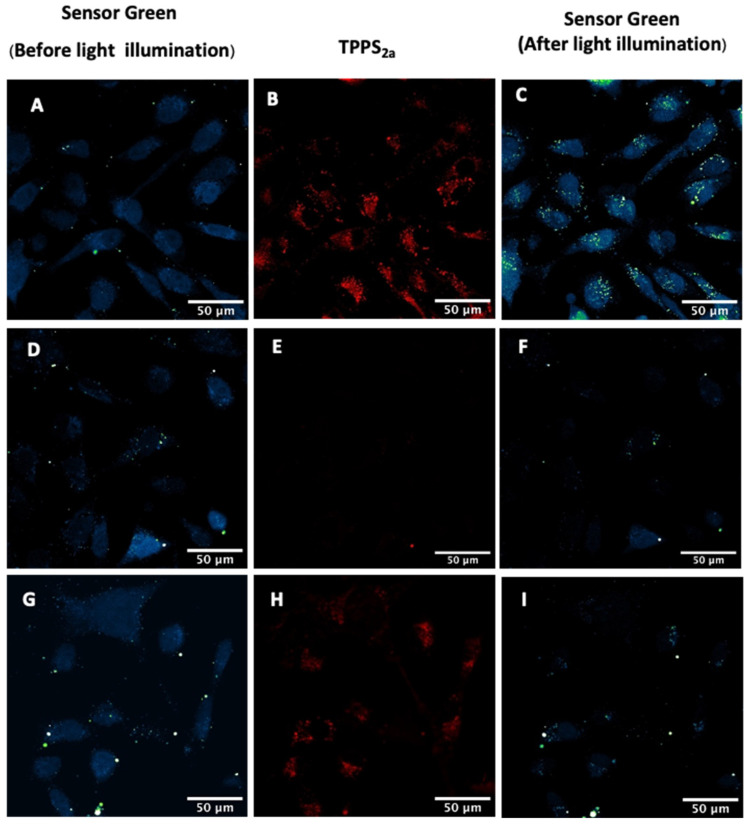
Detection of reactive oxygen species using Sensor Green (SG) following 405 nm illumination. Cells were treated with 1 μM TPPS_2a_ and incubated for 22 h prior to imaging. For the detection experiment, 10 μM SG was added to the cells for a further 2 h prior to imaging, and for the inhibition experiment, 10 μM SG and 25 mM NaN_3_ were added for 2 h prior to imaging. (**A**) SG fluorescence before on-stage illumination; (**B**) TPPS_2a_ fluorescence; (**C**) SG fluorescence after illumination. TPPS_2a_ images are shown in red, fluorescein in false colour (blue, low; green, high). (**D**–**F**): Control sample treated with only SG showing SG fluorescence (**D**,**F**) after illumination using 405 nm laser with no TPPS_2a_ present (**E**). (**G**–**I**): Inhibition of singlet oxygen in co-treated (SG + TPPS_2a_ + NaN_3_) sample, where SG (**G**,**I**) and TPPS_2a_ (**H**) were detected as described above. The scale bar presented in each image is 50 μm.

**Figure 6 ijms-25-04222-f006:**
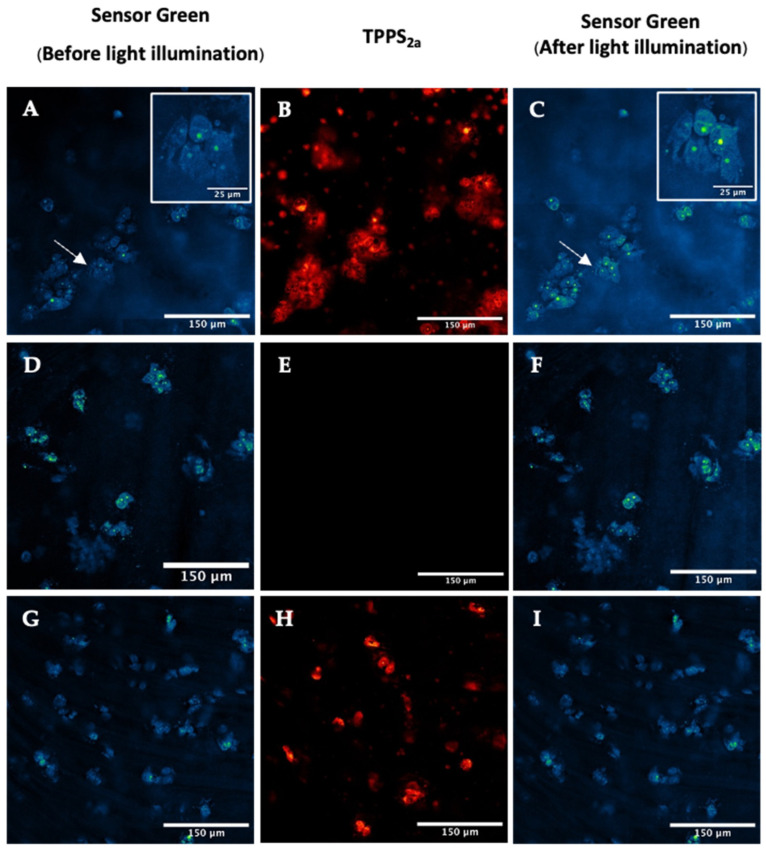
Detection of reactive oxygen species using Sensor Green (SG) in 3D spheroid constructs. Cells were treated with 2 μM TPPS_2a_ and then incubated for 22 h prior to imaging. For the ROS detection experiment, Sensor Green (SG) was added to the cells for a further 2 h prior to imaging, and for the inhibition experiment, Sensor Green and 50 mM NaN_3_ were added for 2 h prior to imaging. (**A**–**C**): Detection of singlet oxygen in co-treated (SG + TPPS_2a_) showing SG fluorescence (**A**) and TPPS_2a_ fluorescence (**B**) prior to 405 nm illumination, (**C**) following 405 nm illumination; (**D**–**F**): Control sample treated with only SG showing SG fluorescence (**D**,**F**) after 405 nm illumination and with no TPPS_2a_ present (**E**). (**G**–**I**): Inhibition of singlet oxygen in co-treated (SG + TPPS_2a_ + NaN_3_) sample, where SG (**G**,**I**) and TPPS_2a_ (**H**) were detected as described above. The scale bar presented in each image is 150 μm. Scale bar for the magnified images highlighted by arrows in insets is 25 μm.

**Figure 7 ijms-25-04222-f007:**
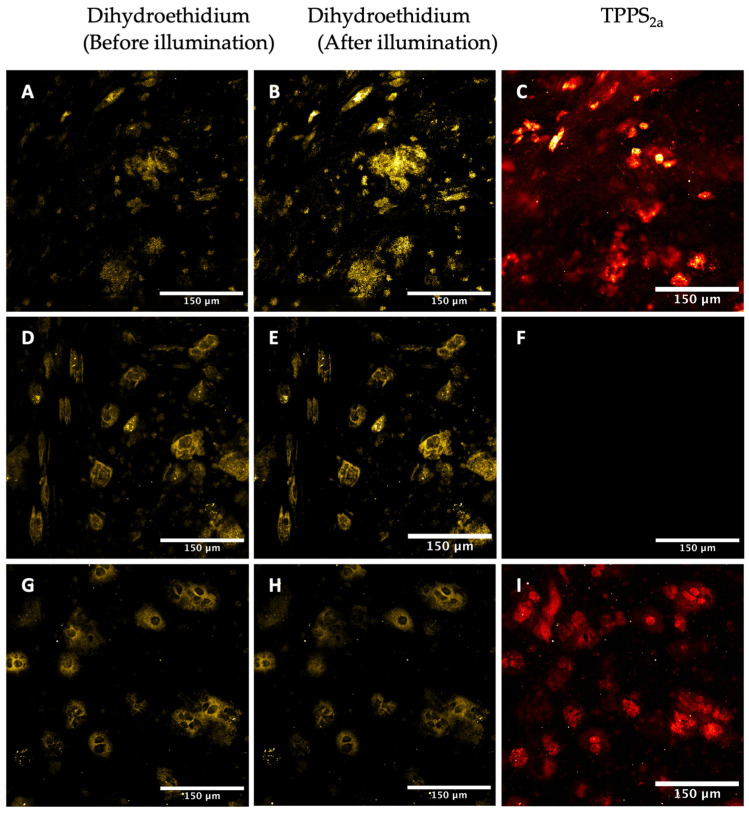
Detection of reactive oxygen species in 3D spheroid constructs using DHE. (**A**–**C**): Cells were incubated with 1 μM TPPS_2a_ for 22 h and 1 μM DHE for a further 2 h, showing DHE fluorescence (**A**) prior to on-stage 405 nm illumination, (**B**) following 638 nm illumination, and TPPS_2a_ fluorescence (**C**). (**D**–**F**): Control images where cells were incubated with only 1 μM DHE and 200 units SOD-PEG for 2 h without TPPS_2a_. (**G**–**I**): Cells were incubated with 1 μM TPPS_2a_ for 20 h, 1 μM DE for a further 2 h, washed, and then 200 units SOD-PEG was added for a further 2 h prior to imaging. The scale bar presented in each image is 150 μm.

**Figure 8 ijms-25-04222-f008:**
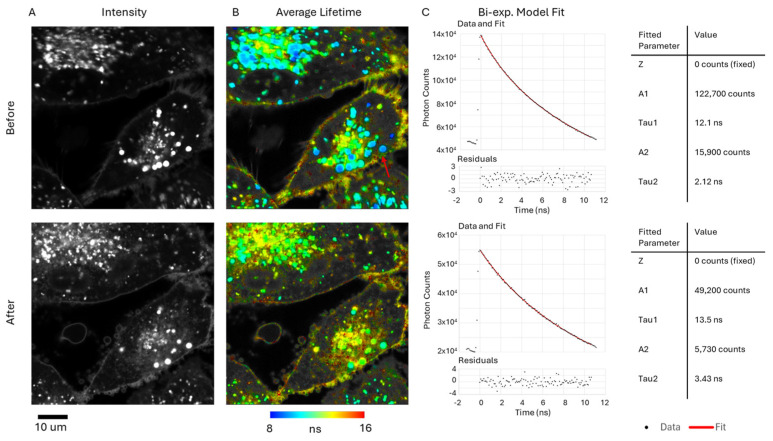
FLIM images of HEY cells in 2D model incubated with 4 μM TPPS_2a_ for 24 h. The top panel shows imaging and lifetime data for initial image recorded. The bottom panel shows the corresponding figures following prolonged on-stage illumination. (**A**) Intensity images; (**B**) FLIM images of cells. Average lifetime determined by mono-exponential fitting is colour-coded onto the intensity image, as shown in the colour bar. (**C**) Bi-exponential fit of all pixels >200 photon counts combined. Arrow highlights a group of punctate vesicles before and after prolonged illumination. The scale bar at bottom left corresponds to 10 μm.

**Figure 9 ijms-25-04222-f009:**
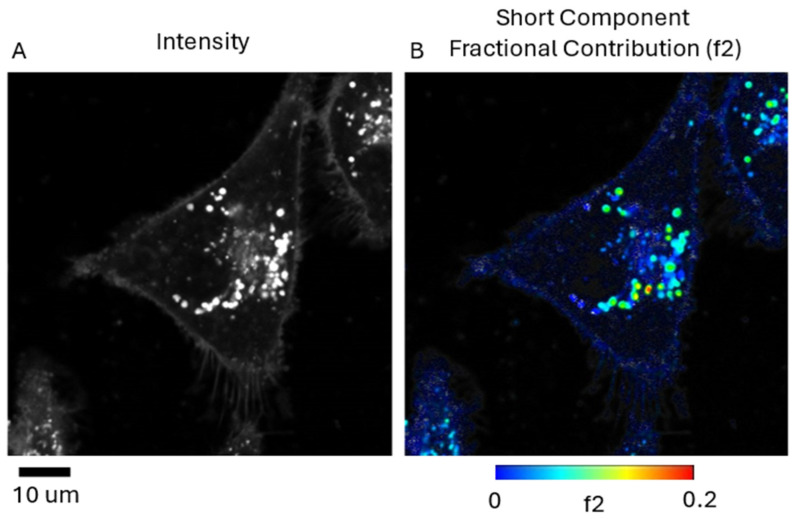
FLIM image of HEY cells in 2D model incubated with 6 μM TPPS_2a_ for 24 h. (**A**) Intensity image; (**B**) fractional contribution (f2) of the short lifetime component determined by global fitting. Global lifetimes detected: 12.9 ns and 2.61 ns. Colour bar shows the scale for the fractional contribution. The scale bar at bottom left corresponds to 10 μm.

**Figure 10 ijms-25-04222-f010:**
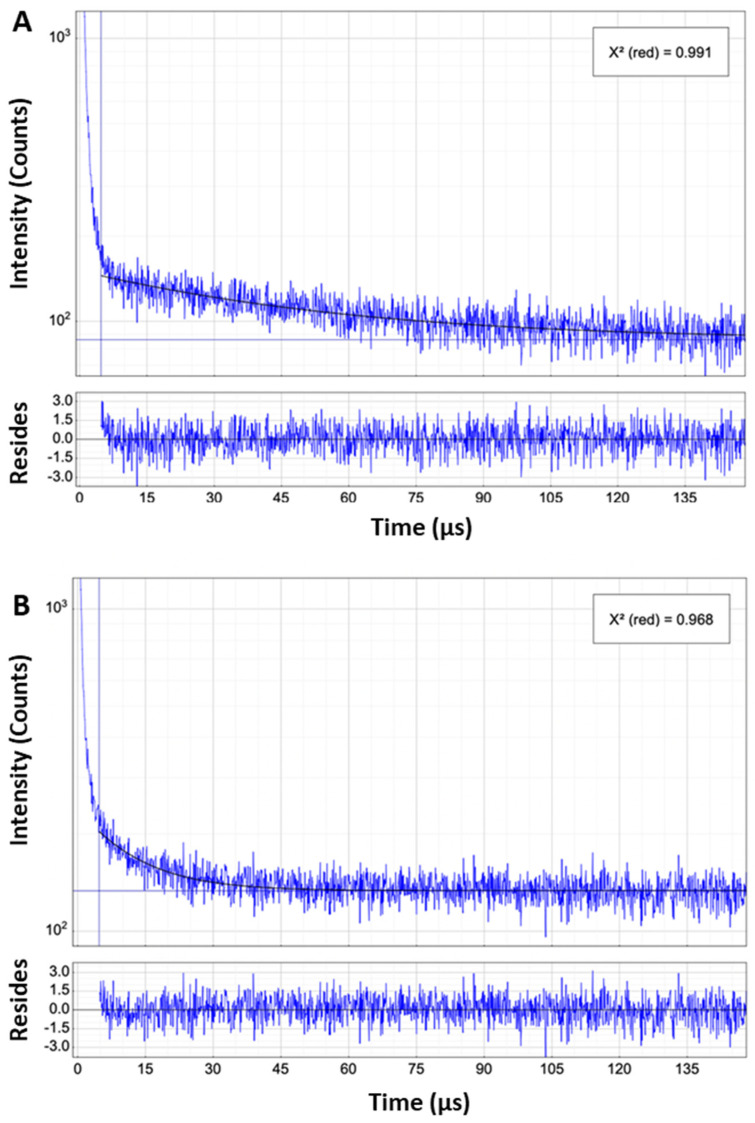
Time-resolved NIR detection of singlet oxygen in 3D spheroid constructs. (**A**) Without addition of azide shown with computed lifetime fit (50 μs) and residuals; (**B**) with addition of azide (0.5 mM) shown with computed lifetime fit and residuals. Initial short-lived signal is from NIR tail of fluorescence. In the presence of azide, which is a physical quencher of singlet oxygen, the long-lived NIR phosphorescence tail is suppressed. The insets in the top right corners show the ‘X’ chi-squared values for the fitting.

## Data Availability

The original contributions presented in the study are included in the article/Appendix A, further inquiries can be directed to the corresponding author/s.

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
