# Peer review of "Cellular Imaging and Time-Domain FLIM Studies of Meso-Tetraphenylporphine Disulfonate as a Photosensitising Agent in 2D and 3D Models"

_ijms, 2024, doi:10.3390/ijms25084222_

Round 1

Reviewer 1 Report

Comments and Suggestions for Authors

The article by Andrea Balukova and colleagues focuses on the cellular imaging and time-domain fluorescence lifetime imaging studies of meso-tetraphenylporphine disulfonate: (TPPS2a) as a photosensitizing agent in both two-dimensional (2D) monolayer cultures and three-dimensional (3D) in vitro cancer models. 

The authors have investigated, the uptake and subcellular localization of TPPS2a and evaluated the photooxidative mechanism using reactive oxygen species (ROS) and lipid peroxidation. The further study dwells into TPPS2a’s photophysical properties in human ovarian cancer HEY cells derived from a peritoneal deposit of a papillary cystadenocarcinoma of the ovary using a time-domain FLIM system with time-correlated single photon counting detection.  

The main question addressed by the authors is how TPPS2a, a photosensitizer, functions in photodynamic therapy (PDT) and photochemical internalization (PCI).

The current study helps to understand the gap in the potential of TPPS2a for PDT and PCI by offering insights into its cellular localization (endolysosomal), phototoxicity mechanism, and the role of light exposure on its aggregation state.

The authors have clearly stated the research question, conducted relevant experiments, and drawn reasonable conclusions, but a few minor things will help to strengthen the manuscript. 

  1. The reference list is comprehensive and includes relevant literature. The authors need to include the most recent publications in this study and discuss the results in the “Discussion” section.

  2. The language is clear and professional throughout the manuscript, but a few minor grammatical errors could be smoothed out. E.g., Section 2.8. Statistical Analysis needs to be the same font formatting. 

  3. The font size in Figure 11, should be increased on the X and Y axis. 

  4. The authors should include the scale bars in the inset figures. (Figure 1, 2, 7).

Author Response

Please see the attachment below:

Reviewer 2 Report

Comments and Suggestions for Authors

The manuscript “Cellular imaging and time-domain FLIM studies of meso-tetra-phenylporphine disulfonate as a photosensitising agent in 2D and 3D models” might be valuable for understanding photodynamic effect in vivo. However, dihydroethidium (DHE) experiments and results should be re-evaluated, and all ambiguities regarding illumination conditions in assays for ROS detection should be removed:

Figures 3, 4 – what was the used laser power and duration of illumination?

Figure 5 – why there is no control with C11 (without porphyrin), before and after 5 min light?

Figure 6 - there are no insets, and there is no white arrow mentioned in caption.

Figure 8 caption – DE should be corrected to DHE (also in SI). The text before says 638 nm wavelength was used for illumination, while here it says 405 nm. Also, above the middle column it says that these images belong to porphyrin fluorescence after illumination, but in the description of figure, it says that this is DHE after illumination. Most importantly, in this experiment, 2-hydroxyethidium, the product of DHE oxidation with superoxide, should be proved (Murphy et al. Nature Metabolism, 2022, 4, 651-662). However, its emission is in red, the same as for porphyrin, so this is problematic and needs to be discussed further. Moreover, some argue that DHE is not a reliable probe for superoxide detection, and that HPLC measurements should be used (Zielonka and Kalyanaraman, Free Radic Biol Med. 2010 15; 48, 983–1001) in addition to confocal microscopy. This must certainly be addressed in the manuscript.

Figure 9 – there is no arrow(s) highlight, which was mentioned in caption.

Lines 679 -  682 – what are the exact wavelengths used for photoactivation of porphyrins (PCI /PDT), with each individual probe, given that what is stated here is not in accordance with the described experiments (red light used only with DHE)?

References 29 and 30 are the same.

Author Response

Please see the attachment below:

Reviewer 3 Report

Comments and Suggestions for Authors

Dear Autors,

In my opinion this manuscript is well prepared. I will ask for better exposition of experimental part. Please show the setup and explain better Singlet Oxygen detection in 3D culture (from preparative side). I think is too many figures from the results, this can be in table ar description. Real new information’s are in the setup (part 2.6) and beautiful results Singlet Oxygen detection what is true challenge in this research .  Please explain how you get value of life time, so readers can learn from this information.

Sincerely

Author Response

Please see the attachment below:

Round 2

Reviewer 2 Report

Comments and Suggestions for Authors

I would like to thank the authors for their answers and additional clarifications. The manuscript has been accordingly improved and I suggest it for publication. I believe this paper will be of interest to a wider readership, especially experts in the field of PDT.

Reviewer 3 Report

Comments and Suggestions for Authors

Thank you